# How trauma related to sex trafficking challenges parenting: Insights from Mexican and Central American survivors in the US

**Marti Marti Castaner**[1]*, **Rachel Fowler**[2], **Cassie Landers**[2], **Lori Cohen**[3], **Manuela Orjuela**[4]

**1** Department of Public Health, Section of Health Research Services, Copenhagen University, Copenhagen, Denmark, **2** The Heilbrunn Department of Population and Family Health, Mailman School of Public Health, Columbia University, New York City, New York, United States of America, **3** ECPAT-USA (End Child Prostitution and Trafficking-USA), New York City, New York, United States of America, **4** Department of Epidemiology, Mailman School of Public Health, Columbia University, New York City, New York, United States of America

* maria.castaner@sund.ku.dk

## Abstract

Sex trafficking, a form of human trafficking for the purpose of commercial sexual exploitation, with a global prevalence of 4.5 million, has pervasive effects in the mental and physical health of survivors. However, little is known about the experiences and needs of Latinx migrants (the majority of sex trafficking victims in the US) after trafficking, particularly regarding parenting. This QUAL-quant study examines how 14 survivors of sex trafficking (mean age = 30) from Mexico and Central America encounter and respond to parenting experiences after escaping sexual exploitation. Combining a bio-ecological model of parenting with Zimmerman's framework on human trafficking we identified how trauma related to sex trafficking can challenge parenting and how relational and contextual pre and post trafficking factors (dis)enable women to respond to such challenges. Psychological consequences of daily victimization primarily manifested in three ways: overprotective parenting in a world perceived to be unsafe, emotional withdraw when struggling with stress and mental health symptoms, and challenges building confidence as mothers. These experiences were accentuated by pre-trafficking experiences of neglect and abuse, forced separation from their older children, poverty post-trafficking, and migration-related stressors. Yet, finding meaning in the birth of their child, having social support, and faith, also enable mothers to cope with such challenges. We conclude that motherhood after surviving sex trafficking presents new challenges and opportunities in the path to recovery from trauma. Interventions at the policy, community and individual level are needed to support survivors of sex trafficking as they enter motherhood.

## Introduction

Sex trafficking, a form of human trafficking conducted for commercial sexual exploitation, is pervasive worldwide [1]. According to estimates from the International Labor Organization, there are approximately 4.5 million victims of sexual exploitation at any given time, and 98%

---

**Data Availability Statement:** All relevant data-participants characteristics and quotes from participants- are available within the paper. Participants did not consent to having the full

---

transcripts made available. Researchers who meet the criteria for access to confidential data, can request additional excerpts related to the study findings that are not within the paper by contacting Columbia University IRB office. Address: CUIMC 154 Haven Avenue, 1st Floor, New York,NY 10032. Phone: 212-305-5883. Fax: 212 305-1316. Email: irboffice@columbia.edu. The study is registered under protocol AAAR9126

**Funding:** MM received discretionary funding from Helena Duch and New York City Department of Health and Mental Hygiene. Founders did not play any role in the study design, data collection, decision to publish, or preparation of this manuscript

**Competing interests:** The authors have declared that no competing interests exist.

are estimated to be women or girls [2]. The United States (US) is one of the top four countries leading the commercial sex markets [3]. While the methods and data used to estimate the prevalence of human trafficking are imprecise and unreliable, available estimates suggest that 15,000–50,000 people are trafficked annually in the US [4]. The largest number of international sex trafficking victims in the United States come from Central America and Mexico (one third are Mexican) and 50% are minors. However, their experiences and needs after trafficking have received limited attention.

Although limited, research on survivors of sex trafficking has identified the needs for mental health services, jobs, life skills training and stable housing, as important in the recovery process. However, little is known about survivors' parenting needs. Data suggest that women surviving trafficking are on average in their late twenties or early thirties when they escape forced prostitution. While there are no estimates about how many trafficked women have children before, during, and after trafficking, many women surviving sex trafficking may form a family by becoming a mother either for the first time or by having another child. Therefore, to provide comprehensive services for female sex trafficking survivors, we must also understand their needs in the context of family formation and explore the impact of sex trafficking on parenting. The present study aims to deepen the understanding of how trauma related to sex trafficking may challenge (or not) parenting during early childhood and explore how relational and contextual pre and post trafficking factors (dis)enable women to respond to such challenges. We focus on the experiences of female survivors trafficked into the US from Mexico and Central America that had children after surviving sex trafficking.

## Impact of sex trafficking on mental health

Female survivors of sex trafficking have experienced multiple forms of trauma, including sexual abuse, psychological abuse, violence, and extortion, often across long periods of time. Reports from media, advocates, and researchers have described the extensive violence that Mexican and Central American women suffer when being trafficked into the US [5, 6]. Women from these regions typically are sold for 15-minute sex acts between 15–20 times per day on a weekday, and 25–35 times per day on a weekend to be considered 'profitable'. There are reports of women being forced to serve a "*doble turno*", two 10 hours shifts consecutively, thereby servicing 60–70 men in a 24-hour period. When women fail to meet their quota, their traffickers frequently beat, starve, and threaten them. Sleep deprivation is the norm for most trafficking victims. Bleeding, vaginal tears and inflammation are common. Additionally, sex buyers frequently subject the women they purchase to strangulation, stabbing, beating and other physical abuse. These acts constitute systematic torture, which may explain the deep trauma many trafficked women suffer [7]. Because of the accumulation of such daily victimization and traumatic experiences, survivors of sex trafficking have high rates of comorbid mental health disorders (PTSD, major depression, anxiety) and health problems [8–10]. For instance, a recent study that interviewed 100 Mexican women that were sex trafficked to work at the US-Mexico border found that 90% had depressive symptoms and 80% thought about committing suicide [5]. A recent review showed that the rates of mental health disorders among sex trafficking survivors worldwide ranged from 30% to 100%. In addition, sex trafficking is more common within marginalized groups and those that lack security and opportunities. A cross-sectional study of human trafficking victims in England found that prior to trafficking, 60% of women had experienced physical violence, and 30% had experienced sexual violence. Similarly, a study among trafficked women and girls in Mexico described how family conflict and separations, poverty, discrimination, and sexual abuse were common among trafficked women and girls, many of which will be moved across the US borders [11].

## Trauma and parenting

A large body of literature has documented the impact of trauma on parenting [12–15]. However, there is a lack of research examining the impact of daily victimizations on the parenting capabilities of women who survived sex trafficking. Parenting, the practices related to the upbringing of a child, ensures the child's health and safety and prepares the child for life as an adult. Parenting hinges on the quality of the parent-child relationship through a range of attitudes, behaviors, and emotional responses towards the child. Responsive and warm parenting behaviors that promote secure and positive parent-child relationships, provide the foundation for the child to develop cognitively, socially and emotionally [16, 17]. The first five years of a child's life are critical for brain development, and as such, early nurturing experiences and relationships with caregivers during this period provide the foundation for later functioning and well-being [18]. However, a parent's history of abuse, trauma and mental health problems may pose a threat to the parent-child relationship [15, 19–21] and in turn, impact the child's development [22].

Prior research on other traumatized groups (victims of child abuse, domestic violence, war, and refugees) suggests that interpersonal trauma can hamper parenting responsiveness [23, 24] and is correlated with lower levels of parenting satisfaction and feelings of parenting efficacy [25]. Higher rates of maternal trauma have been associated with more maternal punitive [12], threatening [26], disengaged [27], and hostile-intrusive behaviors [28]. The literature about posttraumatic stress disorders (PTSD) shows similar trends [29]. A recent systematic review found evidence of a relationship between PTSD and impaired functioning across several parenting domains, including lower parenting satisfaction, increased levels of parenting stress, and more frequent use of hostility and controlling behaviors [15]. Similarly, prior qualitative research suggests that parents suffering PTSD experience parenting challenges as a consequence of their symptoms of avoidance, alterations in arousal and reactivity, and negative alterations of cognitions and mood [30].

Few qualitative studies have addressed the mothering experiences of sex trafficking survivors. Peled and Parker identified the tension between wanting to be "a good mother" by providing their children with better opportunities and feeling like they had "sacrificed their own life" by entering prostitution. In the United States, Busch-Armendariz examined the experiences of international survivors who were seeking family reunification. Survivors were concerned about how to bring their children to the US, how to emotionally re-connect with them and cover the basic material and physical needs [31]. Similarly, Faulkner and colleagues identified the hopes of mothers' survivors of sex trafficking as focusing on the future of their children, reunifying with children in their home country, and finding strength in their faith [32]. These studies provide some important insights into the complexity of motherhood for survivors of sex trafficking and the needs of transnational mothers who have been victims of sex trafficking [33]. However, these studies did not specifically inquire how the traumatic experiences related to trafficking can contribute to parenting challenges alongside other pre and post trafficking factors. Ecological models of parenting suggest that individual (development history, personality, coping resources), interpersonal (marital relationship, social supports) and community-level factors (structural poverty, cultural values) combine in different ways to influence parenting [34, 35]. Therefore, to unpack how complex trauma challenges sex trafficking survivors' ability to build a strong and supportive relationship with their children, we must acknowledge the influence of the broader context in which parenting happens. Women in the current study were both survivors of trafficking and migrants, presenting a new set of challenges (language and cultural barriers, discrimination, fewer opportunities) and potentially intersecting with the burden that traumatic experiences have on women's well-being.

This qualitative study aims to understand how women from Mexico and Central America who were sex trafficked to the United States encounter and navigate parenting a young child born after surviving sex trafficking. Informed by ecological models of parenting, we explore the role that relational and contextual pre and post trafficking factors play in shaping such experiences. By focusing on immigrants from Mexico and Central America we can examine how migration is intertwined with parenting experiences. The results of our analyses will contribute to the development of comprehensive care and services for survivors and their families in order to help them pursue their goals and improve their well-being in the context of motherhood.

## Materials and methods

We employed an embedded design QUALITATIVE-quantitative [36] in which quantitative data about characteristics of sexual exploitation (timing, length, and place) and mothers' current mental state had a secondary role. The study was based primarily on qualitative data from semi-structured interviews with survivors of sex trafficking. Quantitative data served to contextualize qualitative findings and understand differences in parenting experiences among participants.

### Participants

Participants included 14 women from Mexico and the northern triangle of Central America who had been trafficked to the US for commercial sexual exploitation. Participant selection criteria included: being a survivor of sex trafficking from Mexico and Central America, being eighteen years old or older, having a child under 5 years of age at the time of the interview, and being fluent in Spanish or English and therefore able to provide consent in such languages. Exclusion criteria included being in an abusive relationship with an intimate partner at the time of the interview and/or too distressed to participate to avoid re-traumatization. These criteria were based on the observations of case workers attending clients at the non-profit agency from where the subjects were recruited.

Prior to recruitment, we aimed to include at least 12 participants given the expected challenges to negotiate access to survivors of sex trafficking and following prior recommendations about the minimum sample size for qualitative studies [37]. Data collection concluded on pragmatic terms after recruiting 14 participants, determined by the availability of participants and resources to complete the study. Despite this, towards the end of data analysis, new themes did not emerge and existing themes were replicated indicating a level of data saturation. Additional interviews could have added nuances to our analysis or may have uncovered exceptions to the data. Nonetheless, the rich data from 14 participants was sufficient to respond to the aims of this exploratory in-depth investigation.

Participants were recruited through '*Sanctuary for Families*', a nonprofit community agency located in a large city on the East Coast of the United States. The agency provides a range of direct services to survivors of gender-based violence to help them rebuild their lives. It implements an Anti-Trafficking Initiative (ATI), which provides victims of trafficking with legal, counseling, and case management services. Half of the women who receive ATI services are from Latin America.

During July 2018, the agency identified 26 active clients from Mexico and Central America who fit the selection criteria. A case worker, known to all women receiving services, contacted all potential participants to briefly describe the study and ask if they wished to receive further information. If they agreed, the first author (MM) contacted them to discuss the procedures of the study. Of the initial poll of 26 women, two declined to participate, one recently had given

birth and could not participate, and four did not respond. Of the remaining 19 eligible participants, 5 (25%) did not enroll in the study for various reasons. Two did not show up to study appointments twice, two clients experienced scheduling issues, and one participant withdrew because she was experiencing severe PTSD symptoms at the time of the interview.

The first author (MM), who is a native Spanish speaker, invited all potential participants to an individual in-person meeting. At that time, the goal of the study was reviewed again and study consent forms signed. We provided consent forms in Spanish and read them together with all participants to compensate for different literacy levels. We provided participants with multiple opportunities to ask questions prior to providing informed consent and reminded them they could stop the interview at any time. We offered fifty dollars to participants as a token of appreciation for their time, and to compensate for any related costs. After we obtained participants' consent, MM conducted a semi-structured interview and administered a series of surveys. MM conducted all interviews from July 2018 to September 2018.

We obtained ethics approval from the Columbia University Human Research Protection Office and Institutional Review Board (IRB).

## Data collection

Before the interview, we administered a socio-demographic questionnaire (age, marital status, education, employment, income, housing, primary language). Mental health items included past and current use of mental health services. In addition, we administered three screening tools for common mental disorders among survivors of trafficking.

**Mental health symptoms.**　To document the mental state of study participants, we used validated self-administered Spanish versions of the *Patient Health Questionnaire* (PHQ-8) [38], the *Generalized Anxiety Disorder* 7-item scale (GAD-7) [39], and the *Posttraumatic Stress Disorder Check List* (PCL-C) (civilian) [40]. We used PHQ-8 to assess depressive symptoms. PHQ-8 contains items derived from the DSM-IV depression classification system. Respondents are asked to rate the frequency of depressive symptoms in the last 2 weeks on a Likert scale ranging from 0 "not at all" to 3 "nearly every day". Total scores range between 0 and 24 points. We used a cut off $\geq$ 10 to indicate risk for Depression, which has an 88% sensitivity and 88% specificity for major depression, typically representing clinically significant depression [38]. We administered the GAD-7 to assess generalized anxiety symptoms based on some of the DSM-V criteria for General Anxiety Disorder (GAD). Responders are asked to rate the frequency of anxiety symptoms in the last 2 weeks on a Likert scale ranging from 0 "not at all" to 3 "nearly every day". Total scores range between 0 and 21. We used a cut-off point of 10 to indicate risk for GAD, as previous studies that use this cut-off point have a sensitivity and specificity that exceed 80% [39]. We used the PCL-C which measures DSM-IV symptoms of PTSD concerning generic "stressful experiences". This version simplifies assessment based on multiple traumas because symptom endorsements are not attributed to a specific event. Respondents are asked to rate the frequency of PTSD symptoms on a Likert scale ranging from 1 "Not at all" to 5 "Extremely". We used the standard cut-off point of 33 to indicate PTSD [40]. When a participant screened positive for any mental health disorders, we informed caseworkers and they made the necessary referrals to their mental health team for a consultation.

**Semi-structured interview.**　We interviewed all participants in Spanish, their preferred language. One-third also spoke indigenous languages but were fluent in Spanish. The first author, in collaboration with the last author (MO), designed the semi-structured interview to understand participants' experiences and needs after being trafficked, with a particular focus on parenting (see supplemental materials). It included sections that inquired about participant's life before being trafficked, characteristics of trafficking victimization (length of time,

how they entered and exited exploitation), experiences of being pregnant and giving birth, experiences of having a child after surviving trafficking, the relationship with their children and their parenting experience, challenges and support systems, and the impact of being trafficked on their current lives. The semi-structured nature of the interview guide ensured that planned topics were discussed while allowing for the development of unforeseen themes and the discussion of issues relevant to the participant. The ATI director at the referral community agency (LC) reviewed, provided feedback, and approved the interview guide to ensure it was appropriate and respectful to participants' experiences. We piloted the interview guide with two research assistants. MM conducted interviews at the community agency which provided a safe and familiar environment. The agency offered childcare services if needed. Interviews lasted between one and two hours and we recorded them digitally with the permission of participants to allow for subsequent transcriptions and analysis.

## Data analysis

We characterized our participant sample using descriptive analysis of the demographic variables, characteristics of sexual exploitation, and mental health variables. We transcribed all fourteen interviews verbatim in Spanish and entered them into NVivo10 [41].MM and RF, both native Spanish speakers, conducted an iterative data analysis informed by Attride-Stirling's (2001) thematic network analysis [42]. Interviews were analyzed in their original language.

First, MM reviewed all the transcripts and created the first list of codes using an inductive approach. Then RF and MM coded five interviews independently, discussed discrepancies, and refined the coding framework developed by MM. RF continued coding the remaining interviews. When new codes emerged, these were discussed, incorporated in the coding scheme, and used to recode all interviews. This led to 106 codes. In this step, we used Belsky's ecological model of parenting [35] and Zimmerman's framework on human trafficking [10] as guidance to recognize parenting and parent-child relationship experiences after being trafficked, psychological effects of victimization, and ecological factors affecting parenting after being trafficked. Belsky's model inspired us to identify codes that captured individual, interpersonal, and community-level factors that combine in different ways to influence parenting. At the same time, Zimmerman's framework allowed us to capture the temporal component (pre-trafficked vs. post-trafficked) in the experiences of trafficking survivors that influence parenting. Based on these frameworks we selected 65 codes that provided insight into our research question–namely, how women experienced and navigated parenting after their trafficking victimization. Informed by these theoretical frameworks and following thematic network analysis [42] we grouped the 65 codes into 16 basic themes (Table 1). These were then clustered into three organizing themes 1) protecting the child when the world feels unsafe, 2) connecting emotionally with the child: from joy to withdrawal, and 3) regaining control and building confidence as a mother. These three themes were finally connected to the global theme—parenting challenges and opportunities experienced by women after being trafficked.

To ensure data quality, MM and RF discussed and compared all codes, and worked together to identify themes and interpret the findings until agreement was met. This analyst triangulation was a reflexive and iterative process between the two authors to reduce potential bias in data analysis. Moreover, we reviewed data for evidence that could disconfirm the identified themes to make sure that the complexity of participants' narratives was represented [43]. Across this iterative process MO, a pediatrician with extensive expertise and migration and health, and LC, a lawyer with a long career working with survivors of sex trafficking from Latin America and former ATI director at *Sanctuary for Families*, provided critical insight into the context of migrant women who are survivors of sex trafficking.

**Table 1. Thematic network analysis: From basic themes to global theme.**

| Codes | Basic Themes | Organizing theme | Global theme |
|---|---|---|---|
| Fear of being harmed again | Fear after trafficking leads to overprotection | Protecting the child when the world feels unsafe | Parenting challenges and opportunities after trafficking |
| Fear of places | | | |
| Insecurity around other people | | | |
| Fear to move when alone | | | |
| Don't leave the child with others | | | |
| Fear others will harm the child | | | |
| Worried without child | | | |
| Stay at home with the child | | | |
| Avoid people with child | | | |
| Protect children from danger | | | |
| The child can't be separate from me | Overprotection leads to over attachment | | |
| The child cries a lot if I leave | | | |
| Don't have anyone to talk to | (lack of) Social support leads to isolation and only focus on child | | |
| Feel alone | | | |
| Fear of deportation | Acculturative stress leads to isolation and fear | | |
| Communication barriers | | | |
| Lack of English skills | | | |
| Don't understand teachers | | | |
| I feel frozen | Emotional numbness after trafficking | Capacity to Connect Emotionally: from joy to withdrawn | |
| Could not feel connected after birth | | | |
| Desperate when the child cries | Withdraw from child to avoid harsh behaviors | | |
| Step back when can't control child | | | |
| Overwhelmed with child behaviors | | | |
| Joy playing with the child | Capacity to create togetherness and connection | | |
| Cuddle and play | | | |
| Unconditional love | | | |
| Suffer for my child back home | Forced separation from older children lead to ambivalence and emotional overwhelming | | |
| Miss children back home | | | |
| Children back home won't talk | | | |
| My family takes good care of child back home | | | |
| Long reunification process stresses | | | |
| Hope to have my children here | | | |
| Abuse and neglect in childhood | Quality of Childhood experiences influences parenting | | |
| The family did not protect from trafficking | | | |
| Don't know what caring means | | | |
| Caring family | | | |
| Parents are my role model | | | |
| Worried about money daily distracts me from children | Poverty and lack of opportunities create stress leading to emotional disconnect | | |
| Uncertainty about job opportunities | | | |
| Limited job skills | | | |

*(Continued)*

**Table 1.** (Continued)

| Codes | Basic Themes | Organizing theme | Global theme |
|---|---|---|---|
| Insecurity about self | Lack of self-confidence after trafficking | Regaining Control and Building Confidence as a mother | |
| Doubts about telling the child about past | | | |
| Feel incapable | | | |
| Blame if child not ok | | | |
| Feel damaged | | | |
| Shame about past | Judgmental voices enhance low confidence | | |
| Others blame me for my past | | | |
| Others see me damaged | | | |
| Partner supported child in the home country | A supportive partner can restore confidence | | |
| Partner accepted my past | | | |
| My past is secret | | | |
| Shame about past | | | |
| Rely on other mothers | Regain trust in others through shared memberships builds confidence | | |
| Bible helped me to heal | | | |
| The church is like family | | | |
| Church friend give me parenting advice | | | |
| The child is a gift | Focus on the child to regain confidence and build a new self | | |
| Child present from God | | | |
| Child reason to carry on | | | |
| Become a better mother | | | |
| New opportunity to be a mother | | | |
| Nurse help for parenting | Early childhood programs support parenting skills and confidence | | |
| Social work help with parenting | | | |
| Early Head Start teaches me | | | |
| Help see my potential | | | |

## Considerations of researchers' positionality

Throughout this study, authors used reflexivity to reflect on their role as researchers and consider how their background and positionality influenced the study [44]. MM, has a background in clinical psychology and public health and has worked in Latino communities for the last decade. She has an interest in understanding the daily experiences of immigrant families with young children to inform psychosocial interventions that can address social inequalities. She is an immigrant herself and, during the time of the interviews, she had two children under five. She presented herself as such during the interviews. This positioning enabled her to gain the trust of participants and relate to their feelings and experiences of being a new mother. At the same time, she had not experienced sexual exploitation and had a different ethnicity and educational background which allowed her to balance closeness and detachment, and maintain her capacity for critical analysis [45]. MM approached the interviews with empathy and openness yet being aware that participants had been traumatized and therefore they could become emotional during interviews. Her training in clinical psychology allowed her to comfort participants when necessary. In addition, the close collaboration with LC was fundamental to gain access to and trust of participants. LC was the ATI director at *Sanctuary for Families* when the interviews were conducted. Her position, as a Spanish-speaking professional who provided legal counseling to participants and advocated for their rights, enabled MM to gain access to participants. Because

LC was a trusted person, potential participants were open to hearing about the study. In addition, during the interviews, MM was in close communication with LC so that if a participant became mentally distressed, their case workers could follow up with them.

During the analysis phase, MM was aware that her training in clinical psychology and psychoanalysis could lead to 'pathologizing', the act of seeing the participants' behaviors, feelings, and expressions as an indication of a disorder. To counteract MM's clinical background, RF, who is originally from South America, brought a women's rights perspective with her background in public health and work for organizations advocating for women's access to health. Their different backgrounds enabled a rich process of reflexivity throughout the data analysis process in which they reflected upon each other's positionality to enable the trustworthiness of findings. In addition, MO, who is a pediatrician and has worked with immigrant populations in the US and across Mexico was involved in peer debriefings (55) throughout the data collection and analysis process. In addition LC, as an expert in international sex trafficking, contributed with her knowledge about common patterns and characteristics of sex trafficking from Mexico and the Northern Triangle and services available to sex trafficking survivors in the metropolitan New York area. She did not have access to the transcribed interviews to preserve participants' confidentiality, nor was involved in the coding of the data. However, her knowledge enriched the process of interpreting our findings and discussing policy and practice implications.

## Results

### Sample characteristics

Fourteen women aged 20 to 36 years old (mean = 30, SD = 4.22 years) participated in the study. Nearly three-quarters of the participants (73%) came from Mexico (n = 11). All participants spoke Spanish and the majority had been in the US for more than 5 years (mean = 9.13, SD = 3.79). All participants were low income (n = 5) or poor (n = 10) based on income-to-needs calculation, and most participants had a high school education or less. Most of the participants were living with the father of the youngest child (n = 12). All participants had children born in the US. Four also had children prior to trafficking, and four had given birth to children while being trafficked. The children born in the US ranged in age from 2 months to 5 years (mean = 29 months, SD = 16.56 months), and all except one were females. Approximately one-third of participants (n = 5) were "transnational mothers", meaning that they had left children behind in their home country. Detailed demographic data is provided in Table 2.

At the time of the interview, participants had exited trafficking 6.2 years ago on average (range 1.5–14 years). Participants had been forced to enter sex trafficking at an average age of 19 years (range = 14–25), with one third (n = 5) being trafficked as minors. Women from Mexico appeared to be forced into prostitution longer than (mean = 5 years, range 2–10 years) than women from the other countries (range = 2 to 6 months), as they were trafficked through different channels. Participants from Mexico were all forced into trafficking by their "boyfriend". Some participants were trafficked after moving into the home of the "boyfriend," where they learned that their family would be harmed if they refused to be prostituted. Other participants were recruited to the U.S. under false promises of employment opportunities. Those participants from Guatemala and Honduras were forced into sex trafficking by their smugglers after arriving in the U.S. as economic migrants.

### Mental health symptoms

Table 3 shows mean scores for participants on the mental health screening scales, as well as the proportion found to be at-risk using each scale. Only one participant could not complete the assessments. Using standard and validated threshold levels for risk of mental health disorders,

**Table 2. Participant characteristics.**

| Participant | Mother's age | AT Child's age | Country of Birth | Years in the US | Years of SE | Years since SE | Age when trafficked | Exploited in HC | Education | Currently Employed | Num. Children | Children BT/DT | Children living in HC | Living with partner |
|---|---|---|---|---|---|---|---|---|---|---|---|---|---|---|
| A | 27 | 2.8 | G | 5 | 0.4 | 4 | 21 | no | PS | yes | 3 | 2 | yes | yes |
| B | 20 | 2.8 | G | 5 | 0.3 | 4 | 16 | no | SS | no | 1 | - | - | yes |
| C | 28 | 2.0 | M | 8 | 4 | 4 | 19 | yes | SS | yes | 1 | - | - | yes |
| D | 36 | 0.4 | M | 9 | 4 | 6 | 24 | yes | SS | no | 3 | 1 | yes | yes |
| E | 33 | 0.2 | M | 13 | 2 | 11 | 19 | no | SS | no | 4 | 2 | no | no |
| F | 25 | 3 | H | 2 | 0.2 | 2 | 23 | no | VS | yes | 1** | - | - | yes |
| G | 28 | 0.4 | M | 13 | 6 | 7 | 14 | no | PS | no | 1 | - | - | yes |
| H | 29 | 3.2 | M | 12 | 6 | 7 | 15 | yes | Less than PS | no | 3 | 2* | no | yes |
| I | 32 | 2.1 | M | 10 | 5 | 5 | 21 | no | SS | yes | 3 | 2 | yes | yes |
| J | 29 | 1.6 | M | 13 | 10 | 3 | 17 | no | SS | no | 1 | - | - | yes |
| K | 36 | 1.9 | M | 5 | 8 | 5 | 23 | yes | SS | no | 2 | 1* | yes | yes |
| L | 31 | 2.6 | M | 10 | 2 | 9 | 20 | yes | Less than PS | no | 2 | 1* | no | no |
| M | 34 | 1.6 | M | 6 | 4 | 2 | 25 | no | VS | no | 2 | 1* | yes | yes |
| O | 32 | 5 | M | 10 | 2 | 8 | 22 | yes | SS | yes | 1 | - | - | yes |

*Note.* G: Guatemala, M: Mexico, H: Honduras, SE: Sexual exploitation; HC: Home country, AT: After Trafficking; BT: before trafficking, DT

*: during trafficking, PS: finished primary school, SS: finished secondary school, VS: vocational school. Mother's and children's age both refer to their ages at the time of the interview.

**This mother was pregnant at the time of the interview, awaiting a second child. All children were girls except participant's B, who was a boy.

five out of thirteen participants had scores above the threshold for depression (PHQ-8 score >10), three out of thirteen for generalized anxiety (GAD-7 score >10), and ten out of thirteen for PTSD (PCL-C score >33). The four participants that scored below 33 reported mild symptoms (range = 19–28). Although we lacked the statistical power to compare mental health

**Table 3. Distribution of scores on mental health screeners for risk of depression, anxiety, post-traumatic stress disorder.**

| Mental health condition targeted by screener (screening instrument ) | Women with scores above the threshold for increased risk (N)% | Raw score (Mean (Standard Deviation); (range: min, max)) |
|---|---|---|
| Depression (PHQ-8) | - | 7.1 (6.2); (1, 18) |
| score> 10 | 5 (38.5%) | - |
| Anxiety (GAD7) | - | 7.7 (5.0); (2,19) |
| Score > 10 | 3 (23.1%) | - |
| PTSD (PCL-C) | - | 42.3 (15.6); (19,69) |
| score > 33 | 10 (76.7%) | - |

*Note.* PTSD = Post-traumatic Stress disorder. Results based on 13 participants. One participant was unable to complete the surveys dues to a lack of time.

symptom scores between participants from Mexico and those from Central America, only one out of the three participants from Central America had scores above the risk threshold for PTSD, while eight out of ten participants from Mexico had at-risk scores. In total, eleven out of thirteen participants had at-risk scores for any mental health disorder assessed, and four had scores in the at-risk range for more than one disorder. After exiting their trafficking victimization, all but three had received mental health support. However, at the time of the interview, only one participant was receiving mental health treatment.

## Qualitative findings

This section discusses the three organizing themes that constitute the over-arching 'global theme'–namely that survivors of sex trafficking from Latin and Central America experience parenting as a journey with challenges but also opportunities to heal their pain stemming from past victimization. We illustrate how the main three parenting challenges that constitute the organizing themes are connected to the trauma of sex trafficking victimization, and how they are contingent on participants' experiences and relationships before and after the period of being trafficking. In navigating such parenting challenges, women also experienced opportunities to heal their pain, which we highlight across the results.

The first organizing theme '*protecting the child when the world feels unsafe*' describes how mothers, still experiencing generalized fear and lack of trust due to their past victimization, worry about the safety of their children and focus on protecting them above all. In turn, this becomes a challenge as it leads to children becoming overly attached to their mothers. The second theme '*connecting emotionally with the child: from joy to withdrawn*' describes the ambivalence between having a strong emotional connection and unconditional love for their children and challenges in responding and connecting with their children in moments of stress due to the accumulation of daily stressors, childhood experiences of neglect, and separation from older children. The last theme '*regaining control and building confidence as a mother*' captures how past victimization challenges women's self-confidence as mothers and the processes by which women try to fight such insecurity and regain trust in others, which helps them cope with some of their generalized fears.

Below we present the three organizing themes. Through the text we use the words 'some', 'many', and 'most' to represent the typicality and peculiarity of certain patterns in the data; but it is not meant to convey generalizability beyond the study population [46]. 'Some' represents 4–6 out of 14 participants, 'many' represents 7–9 out of 14, and 'most' 10–13 out of 14.

**Protecting the child when the world feels unsafe.** This first theme described how participants experienced parenting with a strong sense of protection, which was fueled by their past traumatizing experiences and the fear that people that trafficked them could try to hurt them again. Even after years of trafficking, participants still experienced flashbacks, anxiety and fear making them feel unsafe. Such fear led to extreme worrying about their children and difficulty being apart from their children which was a parenting challenge, as explained by this mother who survived trafficking 9 years prior to the interview.

> "*Sometimes I have fear for her. You left all that out (trafficking), but then fear comes and you don't feel safe in a place. Because they (children) are also in danger. Now with social media. Sometimes I worry, what if someone finds me and wants to hurt her. . .Sometimes I have had dreams about the person who brought me. I dreamt that he found me and took me with him, not letting me go, telling me that I owe him. (in dreams) I ask for help and no one helps. At that moment I am alone, I don't know what to do, I don't want to leave her alone (daughter). And I wake up very scared (L)*"

Participants' anxieties related to being away from their children were driven by a lack of trust in other people and the fear that something bad could happen to their children, after all the terrible torture they experienced. This lack of trust was described as resulting from the trauma experienced during trafficking, as this participant explained. "*He had me imprisoned, physically and emotionally. . . With everything I have lived, and I have seen in this country, I don't have trust, I have lost it" (H)*. Participants narrated how the lies, humiliation, and violence they suffered, often for a long time, made them distrustful. Many of the participants trafficking experience involved "friends" who tricked them into being sold for sex, or by men with whom they had long-term intimate relationships and, in some cases, children. All participants were economically extorted and most received threats involving harm to their families. Participants also reported that their traffickers had warned that no one would believe them if they sought to report their abuse and that the police would put them in prison and deport them. Deportation was a particularly powerful threat, not only because most participants described being terrified to be in deportation proceedings in the US (incarcerated, abused) but also because they were afraid of suffering retaliation in their country of origin. Such threats served as a strategy to isolate them and keep them from seeking to escape the trafficking ring. This led to further social isolation for both the child and the mother, potentially contributing further to the formation of these over-attached relationships as this participant described.

> "*I don't like to see her sleeping by herself and often I go and stay with her. I don't know, I worry about many things, I worry if I don't feel her close to me. . ..In my mind, it goes that she is missing me, that she is crying. That is why I can't be far from her." (A)*

In turn, most participants also noticed that their children had become too attached to them causing some distress. For example, making them struggle when they had to leave their children with someone else.

> "*It is only she and I. She doesn't see people other than me and the dad. She doesn't socialize with other children. If I leave her, she starts crying. I don't like to leave her alone or with other people in the house.*" (J)

Participants' hypervigilance and avoidance of places and people led them to be over-protective of their children, precluding some participants and their children from socializing and interacting with others in different environments. For example, their fear affected their options for childcare support.

> "*When I had my daughter, I was very closed off, I didn't go out. Because I had been locked up for a long time, I didn't want to be with other people. . .There are many places where we can leave her, but I don't know, I am afraid of leaving her." (H)*

Participants acknowledged the importance of ensuring their children socialized with other children and adults to foster independence. Despite acknowledging the potentially negative consequences of overprotectiveness on a child's health and development, fear and lack of trust in other people prevented some mothers from looking for or pursuing opportunities that would help mitigate these negative impacts. For instance, all participants were entitled to receive childcare support or enroll their children in publicly funded early childcare options, which would provide socializing experiences. However, at the time of the interview, only two had enrolled their children in an Early Education Program and many participants still found it challenging to trust childcare options offered to them. Thus, they delayed bringing their

children to daycare, socializing them with other adults, or even allowing other family members to take care of them, as this participant described:

*"I don't know if it is because of everything that happened to me but I sometimes think that everyone is going to harm my daughter. Especially men. So I have this ugly mentality that sometimes doesn't even let me drop off my daughter at school or something like that. It makes me feel distrustful and not calm when I do leave her." (K)*

Other post-trafficking victimization factors increased the fear that most participants experienced, leading to increased worry and isolation. For example, limited English skills or fear of deportation made the world outside their house and neighborhood more frightening and unpredictable. The fear of deportation of family members, specifically their husbands (or domestic partners), made participants worry about how that would affect themselves and their children emotionally and economically since their partners were the primary household earners. "*The hardest part about being a mother is knowing that my husband could be deported. He is the only one working. I don't work. He has a deportation order. I am worried every day, it is like a "martyrdom"* (G). Combined, these generalized fears increased stress and worry, and contributed to the experiences of parenting under fear that were so clearly articulated during the interviews.

Moreover, family and traditional gender roles combined with limited job opportunities and limited social support placed many participants in a position in which they had to take care of their children alone. Participants recognized that their time spent focusing only on their children made their children too dependent. However, being expected to take care of the children all the time made it harder to break the cycles of overprotection and over-attachment. This pattern made participants, particularly when they had expressed a desire to work, feel overwhelmed.

*"I would like to work, but what can I do? I feel helpless. Because sometimes no one is there. I feel desperate. Our husbands don't value that we are taking care of our children. Taking care of your child is nice but stressful. And when they don't value you, it hurts." (J)*

Together, the fear and lack of trust in others lead participants to be overly attached and overprotective of their children. In addition, their immigrant status, and interrelated aspects such as gender roles, limited job skills, lack of formal education, limited English skills, and poverty precluded most participants from working outside the home. These aspects also reinforced feelings of dependency, a belief that had been internalized after years of trafficking. These factors became persistent barriers to finding opportunities that could help women become more independent from their children and confront their fears.

**Connecting emotionally with their children: From joy to withdrawal.** This theme reflected how participants experienced ambivalence in the emotional connection with their children. Participants described their relationship as an opportunity to love again and develop a strong emotional connection. Even though most participants were still experiencing mental health symptoms, as per the quantitative data, participants described experiencing joy and a deep connection while playing with and caring for their children. As explained by the following participant, this connection strengthened her ability to cope with the emotional consequences of their trauma or/and mental health struggles, *"I feel too happy with her. We always cuddle, play. Sometimes we have bad moments, but she is also like. . . I find shelter with her. She gives me peace." (A)*

At the same time, many participants also experienced feeling emotionally withdrawn from their children when they felt they could not control them (i.e. tantrums, fights between siblings, and anxieties about school). When participants had to take care of their children in moments of stress, negative mood alterations sometimes led them to withdraw emotionally, as described by this participant who struggled with mental health problems after giving birth, '*Everything bothered me. She made a big scene about everything, and I did not pay attention to her. I would get upset with her. Now I understand what was lacking for her. I was not well (L).*'

Similarly, a different participant explained that she would get "*desperate*" when her daughter cried and would tell her to "*shut up, sleep*". She consequently felt as though she was a "*terrible mother*", causing her to withdraw from the situation to avoid being harsh to the child, making her feel unable to respond sensitively to the child's emotions.

Interestingly, participants who described such responses linked their particular moments of feeling emotionally withdrawn from their children to the emotional impact that circumstances beyond the sex trafficking experience had on them. Such circumstances include adverse childhood experiences of neglect or alcoholism in the household, family separation while being trafficked, and post-trafficking victimization daily stressors. Participants described how their own childhood experiences impacted their capacity to connect emotionally with and support their children in moments of stress. Despite some differences in their childhoods, all participants were raised in rural or semi-urban areas with scarce resources and significant material hardship. Six participants described feeling not protected by their caregivers, either because poverty forced their parents to work away from home and not be present, or in some cases, because they experienced abuse and neglect. For all participants with childhood traumatic experiences, the desire to avoid repeating the same pattern and become a better parent was at odds with the lack of positive role models and secure and trustworthy attachment figures.

> '*The most difficult part was how to love them (children). This was hard because I never had much attention or love (from parents) and so this was hard for me. But with time, I started understanding how things were . . . I remember all the suffering when I was a child, and I don't want to do this to my children. And so, this is what makes me be strong and keep going*' (A).

Participants feared recreating the harsh and punitive behaviors they experienced as children and were determined to give their children the love and opportunities they did not have. However, in moments of stress, participants often chose to withdraw from their children as they struggle to respond in sensitive ways and wanted to avoid being harsh. In such cases, the challenges experienced to connect and respond warmly to their children's needs seemed to result from an accumulation of adverse childhood experiences of abuse and neglect, as this participant explained.

> "*I feel that they (parents) never, they couldn't take care of me when I was little. My father was always drunk, always hitting my mother [. . .]I love my mother, but she didn't protect us. . . With my daughter, it is like at the start I didn't know what to do. She was my first daughter too, I had a really bad experience (trafficking), and I wasn't getting better. And at the same time, everything I lived with my family, I don't want to repeat that with my girls. So that is why I say sometimes I am stressed or something, I don't want to even talk to them. Because I feel I am going to yell again.*" (L)

Another factor that made participants feel emotionally absent was forced separation from their older children. This was the case for the five transnational mothers. For all of them, the

trauma of such separation also challenged their capacity to be emotionally present with their younger children. In some ways, the birth of a new child reminded them of their children left in their home country and brought to the surface negative feelings as this participant describes.

> "(when her daughter was born). I was left frozen. Because even though I loved my daughter, I couldn't–How do I tell you? I didn't have any contact with my other son. I felt something was missing. . .. I felt the same love, but no, because your mind is here, but it is lost." (H)

The experience of raising a child born in the US post-trafficking victimization also raised feelings of guilt and sadness for not having provided their older children with the same opportunities. The internal conflict that some participants reported between being happy for having a new child yet sad for those left behind was salient in all transnational mothers. Such ambivalence had tangible emotional impacts on their daily lives and well-being. Transnational mothers experienced challenges being emotionally present when they worried about their older children's safety in their home country, as pimps frequently threatened their families as a means of control. Worries were also related to experiencing feelings of shame; '*will my child find out why I left him*?' These experiences contributed to feeling emotionally divided between "*here*", in the US with their younger children, and "*there*", in their home country with the children left behind.

In addition, the family reunification process, and the relationships with their families of origin either amplified or buffered the impact that forced separations had on the participants' emotional well-being. For transnational mothers, having a good relationship with the family caring for their older children abroad was essential in being able to build a positive relationship with their young children in the US. Knowing their children were safe and well cared for gave them the peace of mind and courage needed to be emotionally present for their younger children. The opposite also held true, where negative familial relationships abroad affected participants' mental state, making them less available for their younger children. For example, a participant explained how her sister was fighting to obtain custody of her child and did not want her to bring the child to the US, affecting her mood and challenging her relationship with her younger children in the US.

> "My sister forbade me to talk to my son. . . I got very upset. And a friend told me that she was not treating him well. . . I don't know how my sister is doing this. . .My child doesn't want to talk to me. I miss him. And this affects me because I am sad and my children (in the US) can feel it. Or they feel it because I don't pay attention to them. . . It is more that I get angry when my daughter is restless" (D)

Moreover, some participants noted how the precarious condition in which they currently lived also challenged their ability to be emotionally present for their children by increasing their stress. All of them were living in low-income households and had limited educational and job opportunities, which in turn limited their earning potential. These systemic barriers prevented participants from fully adapting and succeeding in society, exacerbated existing mental health struggles, increased stress and worries about meeting basic needs, and impacted their ability to be emotionally present for their children.

> "I don't know, hopefully, one day all of that [distrust of others and traumas from trafficking experience] will go away. So that I can really be with them [children] fine. Because sometimes we are fine and sometimes we are not well. But always for the same reason, sometimes I have

*to think about everything, about the rent, the electricity, and when I get it that is when I stress the most." (K)*

In sum, while all participants described positive feelings toward their children and the capacity to embrace moments of joy and emotional connection, the accumulation of experiences of abuse and/or long and uncertain family reunification processes, strained family relationships, and scarcity of resources post-trafficking created a mental burden that sometimes challenged their ability to understand and respond appropriately to the emotional needs of their young children.

**Regaining control and building confidence.** This theme captured participants' experiences moving from feeling insecure as mothers to progressively gaining confidence by redefining their purpose in life by caring for their children. Insecurity was partly a consequence of their post-trafficking mental state, limited support after giving birth, and others' judgmental voices. While motherhood itself entails new challenges and doubts, interpersonal trauma resulting from the trafficking experience affected confidence in their parenting skills. Many participants described instances where they struggled with low self-efficacy. As such, some participants questioned their ability to be a good mother because of what they had suffered.

*"Sometimes I think it would be good if I told her from an early age, or not. Would she understand, or not? Or is it that because of what I have lived through I will not be able to raise her as she should be raised. Will I be able to raise her well? Could I be good for her, or not?" (G)*

The shame associated with having been trafficked contributed to participants questioning their competence in many aspects of their lives, including their role as mothers. This perception about one's capacity to be a "*good mother*" was also influenced by what others said, and how others judged them for having been sexually exploited as these two participants described.

*"My husband told me that I could not care for a daughter. And told him that I was not born knowing how to do it. Because my baby did not want the breast. . . I was not feeling ok with her. I felt bad. I won't be able to care for her, I will be a bad mother, I thought. I always blamed myself." (O)*

*"Sometimes I feel I don't know how to be a mother because I can't control her. . . Because my sister, due to the problem I had (trafficking), they say I am the worst mother, that why do I want to have kids if I don't take care of them." (D)*

As discussed in the examples above, the negative perceptions and comments families made about them affected their confidence and reinforced the disempowerment they had experienced while being trafficked.

However, a range of individual, interpersonal, and social factors enabled participants to deal with these negative thoughts and feelings. For most participants, becoming a mother after surviving sex trafficking was a gift they were grateful for, even when the pregnancy was unplanned. As this participant explains: "*I thank God because after what I went through, I finally have something good (the child). Now I feel some light, after all the darkness. I know there is a lot to do, but I feel as if I had been born again*" (J). After trafficking victimization, many participants believed that they could no longer have children due to both the emotional toll and the physical consequences of trafficking on their reproductive systems. Thus, for all participants, having a child was seen as an opportunity to love again, and for some, it was even seen as a gift from God. All participants described how their children were a source of joy and

brought happiness into their lives after the long journey of psychological suffering, with at least two participants having tried to commit suicide. In many ways, having a child became a way to cope with what had happened to them and helped alleviate some of their mental health symptoms. Many participants expressed that children helped them feel less isolated, motivated them to overcome some of their fears such as going outside, and distracted them from focusing on their negative emotions. As this participant explained, her child was her reason to feel better: *"I feel better because I don't feel so alone anymore. With my daughter, my mind is more occupied and I no longer think so many things like before." (K)*

Similarly, having a child seemed to give participants a reason to carry on, motivating them to recover, to become resilient and to provide their children with the best life possible.

> *"I have to be strong for my daughter. Because now I only think of her. That I would not like anyone to do something like that to her. I have to become strong, with a knot in my throat, I have to endure everything. I mean, to not remember more. . .I am never going to forget it but it is something I don't want to remember. . . .Every day, I'd cry. I would ask God to never again wake me up. Now I ask God to give me many more years of life to see my daughter grow." (E)*

As described in this quote, participants explained how their main goal in life was to provide a nurturing environment for their children. Participants started regaining control of their lives and hope by focusing on their children, giving them unconditional love, and trying to block their memories from the past. During trafficking victimization, all participants had been betrayed, abused and dehumanized. Consequently, their child offered a new opportunity to show and receive unconditional love and feel emotionally connected to someone again. Being able to feel "*full*" of love, as one participant described, helped diminish their feelings of insecurity and helped them feel more capable to be "*good mothers*".

In addition, participants mentioned the importance of family members (past and present), friends (often from their faith communities), supportive partners, and community programs in developing their confidence as mothers. Most participants did not have parents or siblings in the US. In their home countries, their close family would have guided them through the new journey of parenting. Having a positive experience with caregivers growing up helped participants become "*the best parents they could be*" despite the impacts of their trafficking victimization. Stable and caring relationships with their primary caregivers and maintaining transnational relationships enabled them to cope with the parenting challenges they experienced by having someone to turn to when they felt insecure and using their experiences growing up as an example.

> *"My mother gave us a very good example. She took us to church; didn't do bad things. She wanted to be near us. . .I am trying to teach the same values to my daughter. I am trying to be a good mom. I talk often with my mother and my sister and my mother gives me good advice and praise for what I do."(F)*

Besides trying to keep contact with their families back home, finding someone in their community who they felt they could trust, ask questions about day-to-day parenting challenges, and share doubts with brought a sense of relief as this participant explained.

> *'If my mother were here, she would have helped me with my daughter. When she (daughter) was little I did not know anything. . ..[Did you find some support? Someone to talk to about these questions] My husband's niece. She helped me so much. I told her my story and she told*

*me not to worry. She gave me a big hug and told me that she would always help me. She was also alone (in the US). She always tells me if there is something you don't know, ask me. I would call her and tell her how my baby was and she would tell me to do this or do that, "she will be ok," she would say. And I did what she told me because I did not know anything.' (N)*

Similarly, some participants also described how their faith communities played a role in the parenting journey. While sharing concerns with unknown people was very challenging for participants, having a trusted community with whom to share questions and parenting concerns enhanced participants' level of confidence. As this mother described, *"when she was born, it was very hard. She cried and I could not calm her down. It was very stressful. She did not sleep. . . Then, the sisters in the congregation told us that it was ok if the baby cried a little in her bed, "Let her cry!" And I listened to them and she started falling asleep. And now she can sleep in her bed" (G)*. Finding support in, and trusting, their church sisters was part of a healing process that occurred with time. The same participant described her journey from making a friend at church to later trusting her about parenting issues.

*"The girl that lived with me in the shelter took me to a church. In the beginning, it did not help much but then it did. I started looking at life differently and thinking that I had to move forward and that if I was alive, I deserved to live. But I could still not make friends. I did not trust. It was really hard. I had to break that and trust myself first. This took time but I did it. I started trusting myself and then I started trusting others. And until today"(G)*

Likewise, many participants who described a supportive and engaged partner to co-parent, share family responsibilities and support them emotionally, described how it helped them feel better, less stressed, and consequently more available for their children and themselves.

*"My husband has always helped me. In many ways. He has supported me in everything. When I was depressed, when I felt sad, he always talked to me, always said I should keep on going. That "everything at its own time, maybe it can't be so, but you can try again and he [her son] will come". He has always been there with me. In the good and in the bad always with me, and he said to me "I will support you. You will see, your son will come." (H)*

In addition, participants who had received support from structured early childhood programs also described having more confidence in themselves as mothers. Specifically, three participants were engaged in free home visiting programs, with a parenting component at the hospital after birth. Participation in such programs gave them effective parenting strategies and helped them see the potential of their children, thus making them feel more confident raising them.

*"I learned many things. I learned to deal with those problems related to ages two and three, how to cope with her, how to teach her how to grab things. Everything that she is, she is independent. It surprises me because she has such intelligence, potential. (F)"*

In sum, while most participants experienced insecurity about their capacities to care for their children, the belief that their children were gifts and the emotional support or guidance from family (past or present), friends, partners and social institutions enabled them to start building their confidence as mothers. Despite participants' limited social networks, supportive relationships gave them hope that things would get better and encouraged participants to regain agency through motherhood.

## Discussion

This study provides in-depth insight into the parenting challenges and opportunities of fourteen survivors of sex trafficking from Mexico and Central America during early motherhood in the United States. In this sample, parenting challenges included being overly attached to the child, withdrawing emotionally during stressful moments and feeling insecure about parenting capabilities. These challenges were in part influenced by the psychological consequences of the abuse and exploitation suffered during trafficking. However, results also suggested that the complex dynamics of parenting as a survivor of sex trafficking cannot be understood without acknowledging the influence that the pre and post trafficking victimization factors exert on women's mental health and well-being. While the parenting challenges we identified are aligned with prior literature on trauma and parenting, we argue that parenting in the aftermath of sex trafficking is a fluid process in which women experience a complex relationship that fluctuates from supportive and nurturing interactions to overprotective and disconnected interactions. In turn, each of these is influenced by individual, relational, and structural factors across time.

### Parenting challenges in the aftermath of sex trafficking

As the mental health data indicated, most participants in the study were still experiencing PTSD symptoms, despite having escaped their victimization between 2 to 10 years earlier, thus showing the high vulnerability of this group. Most participants noted that their avoidant behaviors and generalized fear often led to overprotecting the young children and missing opportunities for socialization with other children or adults. Thus, feelings of social withdrawal and hypervigilance challenged their parenting capabilities. These results are aligned with prior research on war veterans and traumatized refugees. After traumatic events, people may perceive the world as a dangerous place [24]. Such fear promotes over-attachment and overprotection based on the parents needs to protect their children from feared dangers [23, 47]. When children are overprotected, their independence may be discouraged, hindering the development of their autonomy and possibly leading to anxiety and separation issues [48, 49]. Our results extended this literature by pointing that the particular challenges that immigrant women face such as fear of having family members deported or limited opportunities to find a decent job outside the house may preclude women to stay more at home and accentuate these overprotective patterns.

Participants also discussed how moments of sadness and emotional numbness lead them to feel emotionally disconnected from their children. These challenges are in line with Scheeringa and Zeanah (2001) description of a withdrawn pattern in traumatized mothers [50] and research in war veterans describing how parents experiencing avoidance and withdrawn symptoms can become less emotionally available to their children [30]. However, our results suggest that, based on participants' reflections, it was not the effects of trafficking itself that challenged their capacity to be emotionally present during stressful moments but rather, the accumulation of pre, during and post trafficking victimization stressors. This was especially true for participants that had experienced child abuse growing up. These participants recognized they withdrew from their children as a tactic to avoid repeating the aggressive parenting behaviors they had experienced as a child. Likewise, for mothers who were separated from their older children, the emotional weight of such separation affected their well-being and capacity to be emotionally present for their little children. Therefore, these results may suggest that the compounded trauma of trafficking, forced separations, and prior experiences of abuse and neglect together challenge women's capacity to respond to the emotional needs of their young children in moments of stress.

In addition, our results also suggest that survivors of sex trafficking question their ability to be "*good mothers*". This negative self-perception was directly linked to participants' experiences of disempowerment due to being trafficked for sex. This finding expands prior research showing the negative impact of childhood trauma on parental confidence [25]. In the current study, participants experienced motherhood after trafficking with ambivalence and doubts that were sometimes reinforced by judgmental comments from their social networks. Interpersonal trauma brings profound changes to one's sense of self that can have negative consequences in survivors' self-efficacy [51]. Experiences of continued threat and humiliation by another person may result in feeling unworthy, inferior and more vulnerable [52]. Our results suggest that feelings of inferiority and vulnerability may contribute to the formation of low self-esteem and insecurity related to their ability to be a "*good mother*". Further research should explore the implications of such parenting challenges on children to contribute to the field of transgenerational transmission of trauma.

### The influence of pre and post trafficking factors on women's responses to parenting challenges

Most studies that have examined parenting after traumatic experiences have focused on understanding the relationship between PTSD symptoms and /or trauma, relational patterns, and child outcomes [20, 30]. However, the important role that other contextual factors may have in shaping parenting after trauma has often been overlooked. This study suggests that experiences and relationships before and after trafficking impact parenting by contributing to women's stress levels and ability to cope with the psychological impacts of trafficking. Informed by ecological models of parenting [35], individual, interpersonal, and structural factors help us understand the complexity of parenting after sex trafficking.

At the individual level, participants' narratives about the birth of their child enabled them to navigate the parenting challenges experienced after trafficking and transform such challenges into opportunities. All participants regarded their children as the main reason to live again and a reason to work for a better future which let them experience many moments of joy and playfulness with their children and enjoy a strong emotional connection. This aligns with a growing body of research that has identified the phenomenon of perceived growth following highly stressful experiences, namely how people focus on the positive ways in which their lives have changed as a result of the traumatic experience [32, 53]. Our results show how the process of reestablishing feelings of happiness and trust through motherhood can help women find meaning in life, thus increasing their resilience and ability to cope with the consequences of traumatic experiences [54–56]. Van Ee (2016) called this process the "despite everything" parenting, recognizing the resilience of parents that cope with the mental health sequelae of traumatic experiences yet continue to be sensitive to their children's emotional needs [20]. We argue that for migrant survivors of trafficking when a child is born after escaping victimization, focusing on their child's wellbeing is a core coping mechanism to counteract the psychological impact of their traumatic experience.

Our results suggest that focusing on their child may bring a sense of control over a trafficking survivor's life and working to build a positive relationship with them may fulfill the need to have close and affectionate relationships with others. Despite results showed that mothers struggled with overprotection, focusing on their children may also have the potential of helping women cope with symptoms of avoidance, distrust, and fear by "pushing" them to take small steps to interact with others. For example, by bringing their child to the local playground, asking a godmother to take care of the child for a couple of hours, getting exposed to new environments, and seeking support from people who they have been able to trust such as partners,

church fellows, or family members. At the same time, strengthening relationships with individuals and social support networks can further develop women's confidence, which help women better deal with parenting challenges.

While mothers in our study showed how focusing on their child helped them to cope, attachment theories suggest that if the child becomes "the only thing that makes me happy" to a parent, the parent may turn to the child to have his or her emotional needs met rather than trying to have these needs met by other adults. Such psychological processes could have potentially detrimental effects on the children's development, by challenging the child's attachment and individualization process. At the same time, not fulfilling needs for care, intimacy, and companionship by other adults may also bring more isolation for parents as children grow up [57, 58]. Future observational studies should pay attention to these issues. In addition, while seeing their child as a gift and a new opportunity in life helped block memories from participants' traumatic experiences, this short-term coping behavioral mechanism deserves more attention given the focus of some therapies on recounting the narrative of the traumatic event for long-term well-being [59].

At an interpersonal level, different factors emerged as (dis)enablers of parenting challenges. Findings suggest that women's own childhood experiences with their caregivers influence their perceived ability to parent after trafficking. In particular, the experiences of childhood maltreatment before being trafficked pose an additional challenge to becoming a supportive mother because of the lack of a "role model". It also contributes to a lack of confidence and the tendency to withdraw when the emotional tension between the mother and the child increases to avoid harsh behaviors. This is in line with prior research showing that parents with a story of childhood trauma struggle to meet the emotional demands of parenting while at the same time strive to protect their children against the cycle of abuse [60–62]. Thus, survivors of sex trafficking who are carrying a story of prior abuse and trauma may experience a higher burden and will need additional support [63]. Our findings also complement previous research by suggesting the opposite may also be true. Participants who experienced loving and supportive relationships with their caregivers used that as a foundation to establish positive and nurturing relationships with their children "despite everything" they went through. Therefore, the challenges of parenting after trafficking must be understood in the context of women's own childhood experiences.

Family separations due to trafficking (during and post-trafficking) and the complex relationships with the family of origin and their children born before or during trafficking also affected women's parenting. Our study begins to highlight how separation from children born before or during trafficking intensifies emotional distress and how that affects their emotional availability for their younger children. The guilt and sadness for not being able to raise their children in their home country creates a state of ambivalence towards the child. The issue of family separation in migrant families has received heightened attention recently [64, 65], with many scholars reflecting on the negative emotional impacts on family functioning [66, 67]. In a similar vein, previous literature on sex trafficking survivors has described the emotional toll of being away from their children [31]. While these feelings may be common for many migrants, for survivors of sex trafficking, such ambivalence may be amplified by feelings of shame for their reasons of being away, even if they were forced into the sex trade.

Our findings also suggest that survivors who establish supportive and nurturing relationships with adults–new partner, friends, religious community, and family members in the United States–may be more able to balance their need to protect their children with opportunities to foster independence. This aligns with prior research describing the power of interpersonal relationships in the lives of minor sex trafficking survivors [68]. In the present study, positive relationships with a partner offered both the mother and the child a safe caring

environment that allowed women to re-establish trust and in turn, enable women to be more accepting of help from others [69]. Likewise, finding other women in their family, church, or support groups who were also mothers, and who they felt they could trust, allowed women to share their parenting concerns and feel supported. Trust was described as necessary to develop such positive interpersonal relationships, yet this issue deserves more attention. Halpern-Meekin points out that childhood experiences, cultural norms around social roles, and interpersonal dynamics, come together to influence how people establish high-quality relationships during their life course and when lacking or dysfunctional, can lead to social poverty [70]. With this background in mind, results call for further research to delve into how trust is built and enabled by social structures after sex trafficking and how restorative relationships with caring adults can enhance survivors resilience and parenting [71, 72].

Results also point to the potential role of social institutions in helping women regain confidence and become the mothers they want to be. Early childhood programs that provide trauma-informed care may have the potential to help women build and support nurturing mother-child interactions while reducing parenting challenges [73]. In the study we report here, the three mothers who participated in early childhood programs described very positive experiences of empowerment, gaining confidence, and having a stronger bond with their children. However, most participants in our study were not enrolled in such programs. Our research suggests that generalized fear and lack of trust, coupled with lack of knowledge and cultural norms that encourage women to stay at home and care for their children unaided, play a role in their under usage of available early childhood services. Therefore, more research is needed to understand the barriers that international sex trafficking survivors may experience in accessing motherhood related supportive services that may be beneficial in their recovery process.

Structural aspects of the post-trafficking context also appeared to contribute to the parenting challenges previously described. Results suggest that economic hardship increases women's stress and worry, making them withdraw and feel disconnected from their children. This aligns with an extensive body of research documenting the pervasive effects of poverty on parenting and child development [74, 75]. In our specific population, such hardship is intensified by fears related to not being able to find a job, housing insecurity, deportation, language barriers, and others. These issues are more frequent amongst immigrant survivors of sex trafficking [76] and therefore will require attention from institutions and organizations supporting the recovery process of survivors.

## Implications for policy and practice

Based on the lived experiences of mothers participating in this study, we highlight four specific recommendations to support international survivors of sex trafficking who have become mothers (again) after trafficking. First, survivors of sex trafficking should be offered mental health and psychosocial support during pregnancy and after giving birth, with a focus on the changes that parenting entails. Ensuring that mothers have access to mental health support services around the time of pregnancy and birth is crucial to address previous and ongoing mental health issues that can affect parenting. In the present study, most women (77%) were experiencing risk for PTSD, even after several years (2 to 11) of leaving exploitation, yet only one was receiving mental health support at the time of interviews. Our data showed how fear, worry and negative affect influenced their interactions with children. Best practices should respect women's wishes to revisit (or not) their trafficking experiences and have a dyadic focus, working with both the mother and the child.

Second, results call for the provision of parenting and early childhood education programs that facilitate developing healthy parent-child relationships. These programs need to find the

balance between the mother's desire to be a nurturing parent while at the same time acknowledging the challenges, fear and anxiety resulting from their past victimization. Free home visiting and early childhood programs that understand the role of social networks, including religious and familial, and culture in the recovery process could help women re-build trust. The potentially positive impact of this support was suggested by the positive experience of three mothers and prior literature [77]. These programs may help connect women with other families while also giving their children the opportunity to socialize and support their healthy development.

Third, mental health and parenting support programs will need to be accompanied by measures that support poverty reduction, improve living conditions, provide long term opportunities to regain independence and confidence to accelerate their recovery from trauma [78]. For example, having time to learn English and find training opportunities to seek out better employment opportunities could also help women re-construct their new identity in addition to supporting their role as mothers. Dual generation programs that focus both on providing high-quality early education and provide training opportunities to, for example, learn English, could have the potential to tackle some of these issues [79, 80].

Fourth, particular attention should be given to transnational mothers who have children in their home countries to help them cope with the trauma of separation and prepare them to reunite with their children. Before and after reunification, more support is needed to manage expectations and be prepared to build a relationship with a child with whom they could not initially form an attachment [31, 81]. Such support may be pivotal in the recovery process of women that left children behind and could potentially help them cope with feelings of guilt.

Currently, most anti-trafficking programs are not designed with women's parenting needs in mind. Thus, local and federal guidelines for anti-trafficking programs should explicitly consider the development and implementation of psychosocial support services for pregnant and new mothers and recommend strengthening connections between mothers and home-visiting and early childhood programs.

In addition, we must acknowledge the challenges of implementing any of these support measures for international survivors of sex trafficking. Women's lack of trust and reticence to share their past traumatic experiences could make it challenging to identify those who are in need. Thus, it is crucial first to reduce barriers to access such services by tackling a lack of information, mistrust, and community pressures to stay at home. Two strategies could have the potential to overcome such barriers. First, anti-trafficking agencies and non-profit organizations, such as the one we recruited from, could strengthen their referral pathways to other programs offering psychosocial and parenting support, and early childhood education programs. Referrals coming from a trusted staff in such organizations could be better received. Second, health care providers involved in women's care during pregnancy and after birth could identify women in need of service and link them to free home visiting services, which have been proved to improve the parent-child relationship and child health in vulnerable populations [77]. Research has pointed how prenatal care consultations can be an opportunity to identify victims of trafficking [82]. In addition, there may be benefits in identifying women who already exited sex trafficking but still suffer the consequences of it. A recent study developed and implemented a screening tool during prenatal care with five questions that facilitated the identification of current and past victimization without forcing women to reveal the details of their sexual exploitation [82]. More research is needed to evaluate a scale-up of these screenings and explore how survivors of sex trafficking respond to parenting program offers. In addition, the use of ACES (adverse childhood experiences) screening during prenatal and postnatal care could also facilitate the identification of women who have experienced abuse and neglect before being trafficked and who would benefit from follow-up supports [83].

## Strengths and limitations

A qualitative approach is recommended over a quantitative approach when one is exploring a phenomenon about which little is yet known [84]. In the current study, this approach helped to raise the voices of women who are often silenced and provided a deeper understanding of their experiences. Therefore, the qualitative methodology using individual interviews is an important strength of this study. This exploratory study, while limited in size, has sought to illuminate some of the complexity attached to the parenting experiences of survivors of sex trafficking from Latin America. Our sample was small due to the challenges to access this population, but our methods generated rich data and met qualitative research standards [37]. In addition, we included screening measures of the most common mental health disorders among sex trafficking survivors to help us characterize the current mental state of participants.

This study had several limitations. First, while all participants had been victims of sex trafficking and shared terrifying and traumatic experiences, there was variability in the amount of time trafficked, years since escaping trafficking, and the specific trafficking context. While Mexican participants had been betrayed and forced into prostitution by their boyfriends, two participants from Guatemala and one from Honduras had been trafficked by smugglers with whom they did not have an intimate relationship. These differences could potentially impact their mental health and therefore affect their mothering experience. Despite these differences, our results suggest that all participants experienced parenting challenges to a certain degree, future studies with a larger sample size should consider whether differences in length, timing, and mode of trafficking affect how women experience and navigate parenting after trafficking. Second, the study is based on retrospective qualitative data that may lead to recall bias. However, because we discussed their experiences with pregnancy and motherhood when their children were under five years old, their recall did likely reflect the reality of their experience. Third, our study focused on the experiences of sex trafficking survivors from Central America and Mexico living in a single large city in the host country. Therefore, findings may not be generalizable to other groups of survivors of sex trafficking with a different ethnicity or residing in different parts of the United States. Fourth, all participants were engaged with a community organization that provides support to survivors. Thus, the experiences of victims and survivors who are not receiving any services and support may be different.

## Conclusion

This study highlights how having a (new) child (after trafficking) for immigrant survivors of sex trafficking presents both an opportunity and a challenge in their path to recovery from trauma. The study's findings support the need for holistic community-based models of care to support the wide-ranging needs of immigrant sex trafficked survivors when they become mothers in the host country. With increased collaboration across service providers and agencies and with improved accessibility for referrals to parenting and early education programs, survivors may be able to better access needed supports to assist them to become caring and supportive parents, thus decreasing the likelihood of intergenerational trauma transmission, and improving their abilities to achieve their life goals.

## Supporting information

**S1 Table. Interview guide.**
(PDF)

## Acknowledgments

The authors would like to thank all the participants who contributed their time and effort to the study. We would also like to thank the staff at Sanctuary for Families' Anti-Trafficking Initiative for supporting us to recruit participants and contextualize our findings. The authors would also like to thank Morten Skovdal and Amina Barghadouch for providing feedback on the first manuscript draft and the reviewers for their detailed and constructive feedback.

## Author Contributions

**Conceptualization:** Marti Marti Castaner, Lori Cohen, Manuela Orjuela.

**Data curation:** Marti Marti Castaner, Rachel Fowler.

**Formal analysis:** Marti Marti Castaner, Rachel Fowler.

**Funding acquisition:** Marti Marti Castaner.

**Investigation:** Marti Marti Castaner.

**Methodology:** Marti Marti Castaner, Manuela Orjuela.

**Project administration:** Marti Marti Castaner.

**Resources:** Marti Marti Castaner, Lori Cohen.

**Supervision:** Cassie Landers, Manuela Orjuela.

**Writing – original draft:** Marti Marti Castaner, Rachel Fowler, Cassie Landers, Lori Cohen, Manuela Orjuela.

**Writing – review & editing:** Marti Marti Castaner, Rachel Fowler, Cassie Landers, Lori Cohen, Manuela Orjuela.

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
