## [Decision Letter · Decision Letter 0]

4 Mar 2021

PONE-D-20-34582

How trauma related to sex trafficking challenges parenting: Insights from Mexican and Central American survivors in the US

PLOS ONE

Dear Dr. Marti Castaner,

Thank you for submitting your manuscript to PLOS ONE. After careful consideration, we feel that it has merit but does not fully meet PLOS ONE’s publication criteria as it currently stands. Therefore, we invite you to submit a revised version of the manuscript that addresses the points raised during the review process.

ACADEMIC EDITOR: I appreciated the importance of your work. The reviewers were positive about your manuscript but requested many revisions. You will find that most of the comments are useful to improve the manuscript. Please address all the comments pointed out by the reviewers.

We look forward to receiving your revised manuscript.

Kind regards,

Kenta Matsumura

Academic Editor

PLOS ONE

Journal Requirements:

2. Please include additional information regarding the survey or questionnaire used for the semi-structured interviews in the study and ensure that you have provided sufficient details that others could replicate the analyses. For instance, if you developed a questionnaire as part of this study and it is not under a copyright more restrictive than CC-BY, please include a copy, in both the original language and English, as Supporting Information.

Reviewers' comments:

Reviewer's Responses to Questions

**Comments to the Author**

1. Is the manuscript technically sound, and do the data support the conclusions?

Reviewer #1: Yes

Reviewer #2: No

2. Has the statistical analysis been performed appropriately and rigorously? 

Reviewer #1: N/A

Reviewer #2: Yes

3. Have the authors made all data underlying the findings in their manuscript fully available?

Reviewer #1: Yes

Reviewer #2: No

4. Is the manuscript presented in an intelligible fashion and written in standard English?

Reviewer #1: Yes

Reviewer #2: Yes

5. Review Comments to the Author

Reviewer #1: Overall, this is a thoughtful article on parenting among Latinx woman who have experienced sex trafficking that fills a gap in the literature. A few specifics comments are below.

Abstract: add conclusion/implications. Intro has a lot of facts. Would suggest giving less facts in intro and focus on bigger picture framing and lit gap.

Intro: Good thoughtful lit review for Intro.

Methods: small sample size but good overall description of approach. Considering putting quant data findings in Methods (ie to describe the sample). Did the authors feel they reached saturation on the major themes?

Results: Table 1 formatting is difficult to read. Suggest moving Tables 1 and 2 to Methods since this is framed as qual study. Or frame as mixed methods? Given sample size, it’s stronger as qual.

Qual findings- a bit unclear what are the theme names (bolded text?) versus italics (?subthemes). Within each theme section, define the theme (ie what the term referred to).

Some of the text reads like Discussion, rather than findings (eg pg 24, lines 368-372). Address throughout. Several instances of this occurred.

Discussion: Nicely organized. Excellent connections and comparisons to extant literature. Although this study is exploratory, a bit more drilling done on specific programmatic or policy recs (eg obstetric screening for trafficking?) could strengthen the manuscript to make it more useful.

Reviewer #2: The authors address a critical area - support and experiences of sex trafficking survivors as they enter parenthood. Overall, it is an important study and presents an innovative perspective of survivors. That said, more attention to the discussion of their findings without overstating their recommendations is needed. The themes and stories from your participants get lost in your big picture approach to implications, rather than focusing on what their stories reveal and the very human implications inherent. In part, this may occur because you seem to have identified 'themes' from experiences only reported by a few - which made me wonder if you truly reached saturation and consensus in your sample or if more interviews are needed. This could be helped by adding some 'n' to add context for your statements about 'some participants.' There are quite a few spelling and grammatical errors that would also need to be addressed. I really hope the feedback is helpful to a revision, because I really do believe your study has great merit with those adjustments. More detail for these suggested revisions are included below:

Missing a word perhaps on p. 3 “Despite, methods and data…”

Please add citations to support “…needs for mental health services, jobs, life skills training and stable housing…”

Capitalize ‘trafficking’ in your heading on p.4 and check the remainder of headings for formatting

Remove ‘s’ from ‘which may explains the deep trauma many suffer’

Correct spelling from “80% though about” to ‘thought’

Correct “many of which will moved across” to ‘will move’

Correct ‘and prepare the child for life’ to ‘prepares’ for parallel structure

Please clarify what you mean by ‘maternal frightening (33)’

Add space ‘faith(40)’

Add some context to your inclusion criteria that mothers have a child under age 5. It would make sense that perhaps you were looking at the transition to parenthood OR maybe you were most curious as parent trauma/mental health relate to early child development outcomes (i.e., the critical period of 0 to 6)? Either way, add this to your background literature.

Please clarify how you assessed/screened for inclusion criteria, specifically to know if they were ‘in an abusive relationship’ and under what circumstances would someone have been unable to provide consent. In your description of recruitment and intake this did not seem to come up, so I was curious about the inclusion in the introductory paragraph for this section.

You outline the sections for the interview – I wondered how closely did the actual questions mirror the section titles you provide? The wording of questions can influence the themes we find, so more clarity for this would be helpful.

A statement of researcher positionality would also be helpful to add to trustworthiness of your interpretation of the data. You give some information about MM, but more clear positionality for those involved in coding/interpretation is needed. Additionally, any other measures to add to trustworthiness should be included here (ex: member checking, triangulation, bracketing)

Please expand your description of thematic network analysis to clarify how basic codes, broad categories, basic themes, organizing themes, global themes are related and evolve. This was a little confusing for what was what and how they relate.

Check for consistency in your spacing around statistical data in the text.

The spacing of the table made it hard to read the information included. Did you collect information on the gender of their child? If so, could be interesting to add given the potential for mothers’ experiences of raising daughters to be different than boys given their sexual trauma.

Correct spelling ‘specially their daughters’ to ‘especially’ Note, this statement again reiterated the potential value of adding child gender to your table of participant demographics.

Could you add sample size to clarify the discrepancies between participants who identified “being overly attached and overprotective of their children contrasted with…emotionally withdrawn with their children.” And were there any noticeable trends in the data for which response a parent would provide (e.g., time since their trafficking experience, age of child, gender of child)? There are a few points where you state that “some participants described” which made me wonder if this was a theme overall or if only ‘some,’ then how many of respondents supported this to become a ‘theme’ in your data? For example, of the 5 mothers you identified as being transnational, how many reported the ambivalence you say only “some’ reported? i.e., was there saturation in your data or were more interviews necessary? Likewise, your discussion includes pre-trafficking factors such as family-of-origin abuse as a big takeaway for the pile-up of stressors – but your description of how many/how often this came up is vague (‘some participants felt they were not protected…in some cases, because they were abusive and neglectful’ to then say in your discussion that ‘this was especially true for participants that had experienced child abuse growing up.’)

Do you mean “accomplished” where you say, ‘children helped them feel accompanied’?

Add ‘as’ to ‘overcome some of their fears such AS going outside’

Correct spelling ‘overattachmen’

I also wondered how your questions may have led you to the conclusion that parenting dynamics of ‘survivors of sex trafficking cannot be understood without acknowledging the influence that the pre and post trafficking factors exert…’ I agree this is probably happening and there is a lot of theory (family stress and adaptation theory, ABCX model of stress) to support that, but did your data and participant statements lead you to that conclusion?

I’m a little unclear what you mean by ‘pushing them to seek out networks’ – do you mean that interactions based around their parental role can be helpful to expand social networks and support? There’s a lot of structural barriers to developing social support and social capital – so, if this is your point, then maybe adding a little about those challenges would be good here too (e.g., see Halpern-Meekin’s social poverty work). Likewise, in the discussion you mention survivors who were able to ‘establish supportive and nurturing relationships with adults…may be more able to balance their need to protect their children….’ – did this finding come from your data or are you hypothesizing? I didn’t see a mention of number of social supports or anything like that, so curious where this interpretation came from.

You mention the detrimental effects of the parental role becoming the only source of happiness for a mother for “effects of this behavior on children’s development,” but I also think clarifying the importance of this for the mother’s own well-being is important too.

I’m curious if ‘most participants in our study were not enrolled in such programs’ then how can you make the statement about the importance of these ‘educational and health institutions’ to help mother increase confidence and self-determination? If you are trying to point to other studies for the positive outcomes associated with engagement with such programs then you would need to cite that.

Correct “It is important TO provide more comprehensive maternal…”

Although I agree with your points about the need for trauma-informed care, connect this to your results (specifically, I’m thinking about the descriptive statistics for trauma symptoms even with the longer time since their experience of victimization). Likewise, several of your implications seem to overstate the findings of your study. I think there is a lot you could say with regards to your actual findings with immigrant survivors of sex trafficking rather than big picture ideas that don’t connect as clearly to your results and themes. Also, consider who you are taking to/about, when you call for additional support. You say this could be cross-sectional, but if survivors are hesitant to reach out at all (you said a minority of your sample used services available), where do we start to build this connection? Perhaps with partnership between sites where you recruited and early childcare? Or, for those sites to conduct more follow-up to check on mental health needs or parenting supports? Any implications should directly tie to your results and the theoretical frameworks your outline.

6. PLOS authors have the option to publish the peer review history of their article (what does this mean?). If published, this will include your full peer review and any attached files.

Reviewer #1: No

Reviewer #2: No

---

## [Author Response · Author response to Decision Letter 0]

8 May 2021

Response to reviewers 

Thank you for the opportunity to revise and resubmit our manuscript. 

In the pages below, we detail our responses to the reviewers’ very helpful comments. The manuscript has been edited accordingly. 

Reviewer #1: Overall, this is a thoughtful article on parenting among Latinx woman who have experienced sex trafficking that fills a gap in the literature. A few specifics comments are below.

Abstract: add conclusion/implications. Intro has a lot of facts. Would suggest giving less facts in intro and focus on bigger picture framing and lit gap.

• We appreciate the reviewer’s suggestion, and have substantially revised our abstract in response.

Intro: Good thoughtful lit review for Intro.

Methods: small sample size but good overall description of approach. Considering putting quant data findings in Methods (ie to describe the sample). Did the authors feel they reached saturation on the major themes?

• We have added a paragraph at the beginning of methods section describing the embedded QUAL-quan design of the study. The quantitative results about characteristics of sexual exploitation and current mental health, although secondary, are important to contextualize the qualitative findings. Therefore, we believe is appropriate to have them in the results section. We have now clarifying the design the QUAL-quan and the rationale for the quantitative data collected. 

Results: Table 1 formatting is difficult to read. Suggest moving Tables 1 and 2 to Methods since this is framed as qual study. Or frame as mixed methods? Given sample size, it’s stronger as qual.

• As described above, we have now clarifying the design the QUAL-quan and the rationale for the quantitative data collected. Therefore, we kept the tables in the results section. 

Table 1 is now Table 2. We have modified the formatting to facilitate readability. 

Qual findings- a bit unclear what are the theme names (bolded text?) versus italics (?subthemes). Within each theme section, define the theme (ie what the term referred to).

• Thank you for this suggestion. The italics made reference to basic themes but we realized it creates confusion since the basic themes have longer descriptions. Therefore, we stopped using italics in this section. The basic themes that feed into the organizing themes can now be found in Table 1. This table had previously been in the supplementary data. 

• We have edited the methods and results section to make sure that the organization, presentation, and meaning of each theme is clear. The three organizing themes are the bolded text. 

Some of the text reads like Discussion, rather than findings (eg pg 24, lines 368-372). Address throughout. Several instances of this occurred.

• Thank you for this comment. The text that reads as discussion has been moved into the discussion section. 

Discussion: Nicely organized. Excellent connections and comparisons to extant literature. Although this study is exploratory, a bit more drilling done on specific programmatic or policy recs (eg obstetric screening for trafficking?) could strengthen the manuscript to make it more useful.

• Thank you for this good suggestion. We have now reviewed the section on policy implications and have expanded recommendations based on our findings. 

• In addition, reviewer two thought that in the discussion the links with our results needed to be clearer, particularly regards to recommendations for policy and practice. We have considered the reviewer’s comments and made small edits to make certain that our findings are clearly discussed without overstating our recommendations. 

Reviewer #2: The authors address a critical area - support and experiences of sex trafficking survivors as they enter parenthood. Overall, it is an important study and presents an innovative perspective of survivors. That said, more attention to the discussion of their findings without overstating their recommendations is needed. The themes and stories from your participants get lost in your big picture approach to implications, rather than focusing on what their stories reveal and the very human implications inherent. In part, this may occur because you seem to have identified 'themes' from experiences only reported by a few - which made me wonder if you truly reached saturation and consensus in your sample or if more interviews are needed. This could be helped by adding some 'n' to add context for your statements about 'some participants.' There are quite a few spelling and grammatical errors that would also need to be addressed. I really hope the feedback is helpful to a revision, because I really do believe your study has great merit with those adjustments. More detail for these suggested revisions are included below

• We appreciate Reviewer 2’s comments and have found them most helpful for our revision. We respond to all the comments below

Formatting:

Missing a word perhaps on p. 3 “Despite, methods and data…”

Please add citations to support “…needs for mental health services, jobs, life skills training and stable housing…”

Capitalize ‘trafficking’ in your heading on p.4 and check the remainder of headings for formatting

Remove ‘s’ from ‘which may explains the deep trauma many suffer’

Correct spelling from “80% though about” to ‘thought’

Correct “many of which will moved across” to ‘will move’

Correct ‘and prepare the child for life’ to ‘prepares’ for parallel structure

Please clarify what you mean by ‘maternal frightening (33)’

Add space ‘faith(40)’

• All of these have now been corrected

• ‘Maternal frightening’ has been changed to ‘threatening’ which is more self-explanatory (pag. 5, line 93). 

Add some context to your inclusion criteria that mothers have a child under age 5. It would make sense that perhaps you were looking at the transition to parenthood OR maybe you were most curious as parent trauma/mental health relate to early child development outcomes (i.e., the critical period of 0 to 6)? Either way, add this to your background literature.

• Thank you for pointing that out. We have edited the background section accordingly. (pag. 4, line 84).

Please clarify how you assessed/screened for inclusion criteria, specifically to know if they were ‘in an abusive relationship’ and under what circumstances would someone have been unable to provide consent. In your description of recruitment and intake this did not seem to come up, so I was curious about the inclusion in the introductory paragraph for this section.

• We set as an exclusion criteria that women who did not speak fluent Spanish or English and therefore could not provide consent in such language would not be invited. When the study was conducted, all women that fit the inclusion criteria spoke Spanish. Some also spoke indigenous languages but could communicate fluently in Spanish. Therefore, no women were excluded due to inability to consent in Spanish or English. 

• In addition, to avoid re-traumatization, before a women was referred to study and during the study period, individual case workers serving women at Sanctuary for families were consulted. Case workers were working regularly with potential participants and were aware of their personal circumstances and current mental health state. Therefore, if case workers were aware that potential participants were in an abusive relationship with their current partner or were having a mental health crisis, participants were not invited. This information has been clarified (pag. 7, line 141-149). There was only one case in which the case worker recommended not to invite the participant because she was experiencing PTSD symptoms that were affecting their daily functioning.

You outline the sections for the interview – I wondered how closely did the actual questions mirror the section titles you provide? The wording of questions can influence the themes we find, so more clarity for this would be helpful.

• Thank you for your comment. We have now included the interview guide in the supplemental materials. The questions were more detailed than the titles of the sections. 

A statement of researcher positionality would also be helpful to add to trustworthiness of your interpretation of the data. You give some information about MM, but more clear positionality for those involved in coding/interpretation is needed. 

• We have added a section that describes researchers positionality in the methods section (pag.15)

Additionally, any other measures to add to trustworthiness should be included here (ex: member checking, triangulation, bracketing)

• We have reviewed and edited the methods section to clarify procedures to enhance trustworthiness (pag. 12, line 258-266, and pag. 15, positionality section). First, MM and RF coded 5 interviews independently and reviewed and refined the coding framework developed by MM. In this process discrepancies were discussed until agreement. This was part of analyst triangulation between the two authors that continued as reflexive and iterative process in the organization of themes, and interpretation of findings. In addition, we used reflexivity during the whole research project to reflect upon who preconception and positionality of researchers could influence the results. We describe it in detail in the manuscript. Member checking was not possible for different practical reasons. First, the principal investigator moved outside the US after data was collected and therefore, she could not meet in person with participants to discuss the findings. In addition, we did not have additional funding to compensate participants for a second meeting to conduct member checking. Doing member checking by phone would not have been appropriate given the sensitivity of the topic discussed. Still, we do not think that the lack of member checking invalidates the findings. 

Please expand your description of thematic network analysis to clarify how basic codes, broad categories, basic themes, organizing themes, global themes are related and evolve. This was a little confusing for what was what and how they relate.

• Thank you. We agree that there were some inconsistencies in the description of thematic network analysis. The methods and results have been edited to improve the clarity of it. We have included in the body of the manuscript the supplemental table showing the thematic network analysis: from basic themes to global theme as part of the results section (pag.15). In addition, at the beginning of the results section we added a summarizing paragraph describing how the three organizing themes create a thematic network –the global theme (pag. 20, line 354-378).

Check for consistency in your spacing around statistical data in the text. The spacing of the table made it hard to read the information included. 

• We agree and apologize and have reformatted both prior tables (now tables 2 and 3) to make them easier to read. 

Did you collect information on the gender of their child? If so, could be interesting to add given the potential for mothers’ experiences of raising daughters to be different than boys given their sexual trauma.

• Yes. This information has been added. All but one child born after trafficking were females, thus we were unable to examine whether experiences differed by the child’s gender

Correct spelling ‘especially their daughters’ to ‘especially’ Note, this statement again reiterated the potential value of adding child gender to your table of participant demographics.

• This spelling error has been corrected, and as mentioned above.

Could you add sample size to clarify the discrepancies between participants who identified “being overly attached and overprotective of their children contrasted with…emotionally withdrawn with their children.” Were there any noticeable trends in the data for which response a parent would provide (e.g., time since their trafficking experience, age of child, gender of child)? There are a few points where you state that “some participants described” which made me wonder if this was a theme overall or if only ‘some,’ then how many of respondents supported this to become a ‘theme’ in your data? 

• Thank you for this helpful comment. We realized that the way in which we wrote the results may have lead the reader to think that mothers either experienced being over-attached or withdrawn, or that organizing themes were not representative of all participants’ experience. We have reviewed the results section and clarified these issues. The three organizing themes reflected the experiences of all mothers. In contrast, basic themes did not necessarily capture the experience of all participants. The basic themes allowed use to find interesting patterns within the organizing themes. An example is that mothers who expressed having had traumatic experiences during their childhood were the ones describing more instances in which they struggled to connect emotionally with their post trafficking children and also becoming withdrawn to avoid being too harsh. 

• Regarding the organizing themes, all mothers described their need to protect their children fueled by fear as a consequence of past victimization. Regarding the second theme - Capacity to connect: from joy to withdrawn- we have edited the text to reflect that the commonality among all mothers was their desire and capacity to connect emotionally with their children and how different circumstance lead particular groups of mothers to struggle more in stressful situations. We describe in more detail (and include indicators of frequency or sample size when needed as suggested) trends in the data that describe what leads some mothers to withdraw from their children in certain stressful situations. We also emphasize that is not that mothers are in s state of withdrawal from their children. Instead, is during moments of stress or perceived lack of control that they respond in such way. For example, all five transnational mothers described how they felt disconnected and withdrawn from their children when they were thinking about their children back home and worried about them. Or 42% of mothers that described neglect and abuse in childhood also described challenges to respond in a sensitive manner to their children’s distress that lead to withdrawn behaviors. 

• As Sandelowski (2001) recommends, we have included a statement that explains the meaning that we ascribe to the terms ‘some’, ‘many’, and ‘most’ in the results section. When using these words to illuminate the intensity of certain patterns in the data, we have added more context to ensure that readers understand the meaning of these ‘quantitative’ words. (pag.21, line 376)

Sandelowski, M. (2001). Real qualitative researchers do not count: the use of numbers in qualitative research. Research in nursing & health, 24(3), 230-240.

For example, of the 5 mothers you identified as being transnational, how many reported the ambivalence you say only “some’ reported? i.e., was there saturation in your data or were more interviews necessary? 

• Thank you for this question. The use of some has created confusion. As we described in the text, the ambivalence was present in all transnational mothers ‘The internal conflict that some participants reported between being happy for having a new child yet sad for those left behind was salient in all transnational mothers’. We used ‘some’ given that transnational mothers were 5. We now state all 5 transnational mothers to avoid confusion.

• Regarding data saturation, we have now included a statement to clarify this issue. It is recommended that qualitative studies have a minimum sample size of 12 to reach data saturation (Clarke & Braun, 2013; Fugard & Potts, 2014; Guest, Bunce, & Johnson, 2006). Therefore, we set to include at least 12 participants given the expected challenges to recruit survivors of sex trafficking. Data collection concluded on pragmatic terms determined by the availability of respondents at Sanctuary for Families and resources to complete the study. Despite this, towards the end of data analysis new themes were not emerging and the three organizing themes were being replicated indicating a level of completeness. We acknowledge that additional interviews could have added nuances to our analysis or may have uncovered exceptions to the data. Nonetheless, the rich data from 14 interviews was sufficient to respond the aims of this small in-depth investigation. 

Likewise, your discussion includes pre-trafficking factors such as family-of-origin abuse as a big takeaway for the pile-up of stressors – but your description of how many/how often this came up is vague (‘some participants felt they were not protected…in some cases, because they were abusive and neglectful’ to then say in your discussion that ‘this was especially true for participants that had experienced child abuse growing up.’)

• Thank you. We have now clarified this and included the N of mothers that described abuse and neglect pre-trafficking. Six mothers (42%) described experiences of abuse and neglect when they were children. 

Do you mean “accomplished” where you say, ‘children helped them feel accompanied’?

• We meant less lonely, less isolation. This was translated from a Spanish term ‘acompanadas’ that mothers used when describing their relationship with their children. We have now changed the language to avoid confusions. 

Add ‘as’ to ‘overcome some of their fears such AS going outside’

• done

Correct spelling ‘overattachmen’

• done

I also wondered how your questions may have led you to the conclusion that parenting dynamics of ‘survivors of sex trafficking cannot be understood without acknowledging the influence that the pre and post trafficking factors exert…’ I agree this is probably happening and there is a lot of theory (family stress and adaptation theory, ABCX model of stress) to support that, but did your data and participant statements lead you to that conclusion?

• Thank you for pointing out this issue. The data and participant statement showed the influence of pre and post trafficking experiences in their parenting. We have reviewed the results and quotes selected, in order to be certain that they reflected the links that were evident in the data, as can be seen in the basic themes (Table 1). When necessary we have added additional quotes to demonstrate the connection between pre and post trafficking experiences and parenting. For example, regarding the influence that pre-trafficking, and particularly early adverse experiences exert on their parenting challenges:

‘The most difficult part was how to love them (children). This was hard because I never had much attention or love from my parents and so this was hard for me. But with time, I started understanding how things were … I remember all the suffering when I was a child, and I don’t want to do this to my children. And so, this is what makes me to be strong and keep going’ (A).

I’m a little unclear what you mean by ‘pushing them to seek out networks’ – do you mean that interactions based around their parental role can be helpful to expand social networks and support? There’s a lot of structural barriers to developing social support and social capital – so, if this is your point, then maybe adding a little about those challenges would be good here too (e.g., see Halpern-Meekin’s social poverty work). 

Likewise, in the discussion you mention survivors who were able to ‘establish supportive and nurturing relationships with adults…may be more able to balance their need to protect their children….’ – did this finding come from your data or are you hypothesizing? I didn’t see a mention of number of social supports or anything like that, so curious where this interpretation came from.

• This statement comes from the data, however, because we only described the role of the partner and family briefly within the third theme, we recognize that it might benefit from additional clarification. We have expanded this section within the third organizing theme which stems from basic themes presented in Table 1. We now expand the role of church and female relatives in the lives of women post trafficking and what these mean for women in terms of parenting. We also describe the role that trust plays in creating such relationships. 

• Thank you for making us consider Halpern-Meekin work about social poverty. It is indeed very relevant for our study. This paper does not focus specifically on the barriers to develop social capital and social support. However, we acknowledge that this may be an issue since participants indeed described limited social networks. Therefore, we highlight Halpern-Meekin ideas in the discussion and call for further investigation for further research to delve into how trust is built and enabled by social structures after sex trafficking. (pag. 43-44, line 859-864)

You mention the detrimental effects of the parental role becoming the only source of happiness for a mother for “effects of this behavior on children’s development,” but I also think clarifying the importance of this for the mother’s own well-being is important too.

• We have expanded the discussion to consider this important point.

I’m curious if ‘most participants in our study were not enrolled in such programs’ then how can you make the statement about the importance of these ‘educational and health institutions’ to help mother increase confidence and self-determination? If you are trying to point to other studies for the positive outcomes associated with engagement with such programs then you would need to cite that.

• The three mothers that were engaged in early childhood programs discuss how beneficial those were in gaining confidence as mothers and having a supportive relationship with their children as it was described in the results. However, we understand that with only three mothers participating in such programs we need to be cautious . A citation has been added pointing to other studies that suggest the role of ECE programs and two generation programs for underserved and vulnerable populations. 

Correct “It is important TO provide more comprehensive maternal…”

• done

Although I agree with your points about the need for trauma-informed care, connect this to your results (specifically, I’m thinking about the descriptive statistics for trauma symptoms even with the longer time since their experience of victimization). 

• Thank you for this important suggestion. We have connected this point to our discussion more clearly

Likewise, several of your implications seem to overstate the findings of your study. I think there is a lot you could say with regards to your actual findings with immigrant survivors of sex trafficking rather than big picture ideas that don’t connect as clearly to your results and themes. 

• Thank you for this constructive comment. We have rewritten the implications section accordingly to clearly link the discussion to our findings and state 4 clear program and policy implications-

Also, consider who you are taking to/about, when you call for additional support. You say this could be cross-sectional, but if survivors are hesitant to reach out at all (you said a minority of your sample used services available), where do we start to build this connection? Perhaps with partnership between sites where you recruited and early childcare? Or, for those sites to conduct more follow-up to check on mental health needs or parenting supports? Any implications should directly tie to your results and the theoretical frameworks your outline.

• Thanks you for this good suggestion. We have reviewed the implications section and focus on concrete and tangible recommendations linked to our findings. We highlight the important role in anti-trafficking agencies and NGO’s to link survivors with parenting and early childhood programs and mental health services that take a dyadic approach.

---

## [Editor Report · Decision Letter 1]

19 May 2021

How trauma related to sex trafficking challenges parenting: Insights from Mexican and Central American survivors in the US

PONE-D-20-34582R1

Dear Dr. Marti Castaner,

We’re pleased to inform you that your manuscript has been judged scientifically suitable for publication and will be formally accepted for publication once it meets all outstanding technical requirements.

Kind regards,

Kenta Matsumura

Academic Editor

PLOS ONE

Additional Editor Comments:

The authors addressed all the comments pointed out by the reviewers, thank you.
---

## [Editor Report · Acceptance letter]

24 May 2021

PONE-D-20-34582R1 

How trauma related to sex trafficking challenges parenting: Insights from Mexican and Central American survivors in the US 

Dear Dr. Marti Castaner:

I'm pleased to inform you that your manuscript has been deemed suitable for publication in PLOS ONE. Congratulations! Your manuscript is now with our production department. 

Kind regards, 

on behalf of

Dr. Kenta Matsumura 

Academic Editor

PLOS ONE